# A Large-Scale Database for Graph Representation Learning

**Scott Freitas**
Georgia Institute of Technology
safreita@gatech.edu

**Yuxiao Dong***
Microsoft Research
ericdongyx@gmail.com

**Joshua Neil**
Microsoft ATP
joshua.neil@microsoft.com

**Duen Horng Chau**
Georgia Institute of Technology
polo@gatech.edu

## Abstract

With the rapid emergence of graph representation learning, the construction of new large-scale datasets is necessary to distinguish model capabilities and accurately assess the strengths and weaknesses of each technique. By carefully analyzing existing graph databases, we identify 3 critical components important for advancing the field of graph representation learning: (1) large graphs, (2) many graphs, and (3) class diversity. To date, no single graph database offers all these desired properties. We introduce MALNET, the largest public graph database ever constructed, representing a large-scale ontology of malicious software function call graphs. MALNET contains over 1.2 million graphs, averaging over 15k nodes and 35k edges per graph, across a hierarchy of 47 types and 696 families. Compared to the popular REDDIT-12K database, MALNET offers **105× more graphs**, **39× larger graphs** on average, and **63× more classes**. We provide a detailed analysis of MALNET, discussing its properties and provenance, along with the evaluation of state-of-the-art machine learning and graph neural network techniques. The unprecedented scale and diversity of MALNET offers exciting opportunities to advance the frontiers of graph representation learning—enabling new discoveries and research into imbalanced classification, explainability and the impact of class hardness. The database is publicly available at www.mal-net.org.

## 1 Introduction

The emergence of graph data across many scientific fields has led to intense interest in the development of representation learning techniques that encode structured information into low dimensional space for a variety of important downstream tasks (e.g., toxic molecule detection, community clustering, malware detection). However, recent research focusing on developing graph kernels, neural networks and spectral methods to capture graph topology has revealed a number of shortcomings of existing benchmark datasets [1, 2, 3, 4], which often contain graphs that are relatively: (1) limited in number; (2) smaller in scale in terms of nodes and edges; and (3) restricted in class diversity. The state

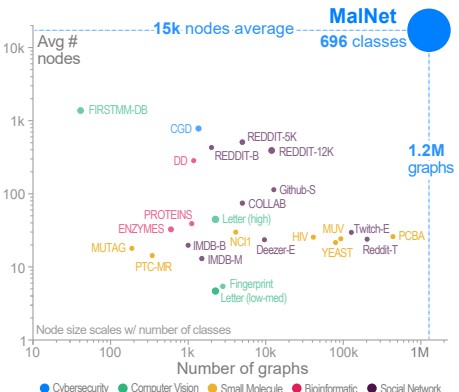

Figure 1: MALNET has 1.2M graphs averaging 15k nodes and 35k edges per graph.

---

*Now at Meta AI and work done when at Microsoft.

35th Conference on Neural Information Processing Systems (NeurIPS 2021) Track on Datasets and Benchmarks.

of graph representation benchmarks (e.g., PROTEINS [5], IMDB [6], REDDIT [6]) is analogous to MNIST [7] at its height—a staple of the computer vision community, and often the first dataset researchers would evaluate their methods on. The graph representation community is at a similar inflection point, as it is increasingly difficult for current databases to characterize and differentiate modern graph representation techniques [1, 2].

To address these issues, we introduce a new graph database called MALNET, a large-scale ontology of malicious software function call graphs (FCGs). Each FCG represents calling relationships between functions in a program, where nodes are functions and edges indicate inter-procedural calls. Through MALNET, we make three major contributions:

- **MALNET: Largest Database for Graph Representation Learning.** MALNET contains 1.2 million function call graphs, averaging over 15k nodes and 35k edges per graph, across a hierarchy of 47 types and 696 families (Figure 1). This makes MALNET the largest public graph database constructed to date, offering **105× more graphs**, **39× larger graphs** on average, and **63× more classes** compared to the popular REDDIT-12K database. We release MALNET with a CC-BY license, allowing users to share and adapt the database for any type of use. We also provide code on Github: `https://github.com/safreita1/malnet-graph`.

- **Revealing New Discoveries.** The unprecedented scale of MALNET enables new and important discoveries that were previously not possible. Leveraging the function call graphs in MALNET, we study popular graph representation learning techniques in depth, and reveal: (1) the significant challenges they face in terms of scalability and their ability to handle large class imbalance and (2) that simple baselines can be surprisingly effective at the scale of MALNET;

- **Enabling New Research Directions.** MALNET offers unique opportunities to advance the frontiers of graph representation learning by enabling research into *imbalanced classification*, *explainability* and the impact of *class hardness*. We believe the diversity, scale and natural imbalance of MALNET will enable it to become a benchmark dataset to meet the future research needs of the graph representation community. By open-sourcing MALNET, we hope to inspire and invite more researchers to contribute to this exciting new resource.

## 2 Properties of MalNet

We begin by analyzing 5 key properties of the MALNET database—(1) *scale* (number of graphs, average graph size), (2) class *hierarchy* (3) class *diversity*, (4) *class imbalance* and (5) *cybersecurity applications*. In Section 2.1 we compare MALNET against common graph classification datasets, summarizing the differences in Table 2.

**Scale.** MALNET contains 1,262,024 function call graphs across 47 types and 696 families of malware. When stored on disk, MALNET takes over 443 GB of space in edge list format, with each graph containing 15,378 nodes and 35,167 edges, on average. This makes MALNET the **largest public graph dataset constructed** to date in terms of *number of graphs*, *average graph size* and *number of classes*. In Table 1, we provide descriptive statistics on the number of nodes, edges, and average degree of ten of the largest graph types (see Appendix Table 4 for a full comparison). We believe that this scale of data is crucial to the future development of graph representation techniques as current databases are too small to effectively differentiate and benchmark techniques on non-attributed graphs [1, 2, 3, 4].

**Hierarchy.** Function call graphs are assigned a general *type* (e.g., Worm) and specialized *family* label (e.g., Spybot) using the Euphony [8] classification structure (see Figure 2). To generate these labels, Euphony takes a Virus-Total [9] report containing up to 70 labels across a variety of antivirus vendors and unifies the labeling process by learning the patterns, structure and lexicon of vendors over time. While Euphony provides state-of-the-art per-

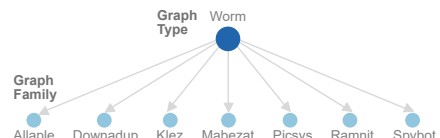

Figure 2: Example of the graph type "worm" and its 7 families.

formance, this task is considered an open-challenge due to both naming disagreements [10, 11] and a lack of adopted naming standards [8] across vendors. To help address this issue, we collect and release the raw VirusTotal reports containing up to 70 antivirus labels for each graph.

| Type | # graph | # fams. | Nodes | | | | Edges | | | | Avg. Degree | | | |
|---|---|---|---|---|---|---|---|---|---|---|---|---|---|---|
| | | | min | mean | max | std | min | mean | max | std | min | mean | max | std |
| Adware | 884K | 250 | 7 | 14K | 211K | 16K | 4 | 31K | 605K | 38K | 0.50 | 2.21 | 6.24 | 0.36 |
| Trojan | 179K | 441 | 5 | 15K | 228K | 18K | 4 | 34K | 530K | 42K | 0.58 | 2.05 | 6.74 | 0.52 |
| Benign | 79K | 1 | 5 | 35K | 552K | 30K | 3 | 79K | 2M | 74K | 0.58 | 2.13 | 5.30 | 0.31 |
| Riskware | 32K | 107 | 5 | 12K | 173K | 16K | 4 | 30K | 334K | 39K | 0.58 | 2.16 | 5.42 | 0.56 |
| Addisplay | 17K | 38 | 37 | 13K | 98K | 15K | 37 | 28K | 246K | 34K | 0.92 | 1.97 | 4.38 | 0.37 |
| Spr | 14K | 46 | 12 | 28K | 169K | 21K | 7 | 67K | 369K | 52K | 0.58 | 2.27 | 4.70 | 0.44 |
| Spyware | 7K | 19 | 12 | 5K | 55K | 6K | 7 | 11K | 121K | 14K | 0.58 | 1.95 | 4.27 | 0.46 |
| Exploit | 6K | 13 | 19 | 24K | 102K | 14K | 14 | 45K | 250K | 30K | 0.74 | 1.88 | 3.34 | 0.33 |
| Downloader | 5K | 7 | 37 | 20K | 107K | 28K | 37 | 46K | 321K | 63K | 0.96 | 1.68 | 3.53 | 0.66 |
| Smssend++Trojan | 4K | 25 | 16 | 34K | 147K | 19K | 13 | 82K | 387K | 48K | 0.81 | 2.39 | 3.78 | 0.23 |

Table 1: Descriptive statistics for 10 largest graph types. See Appendix Table 4 for all graph statistics.

**Diversity & Imbalance.** MALNET offers 47 types and 696 families of function call graphs following a long tailed distribution with imbalance ratios of $7,827\times$ and $16,901\times$, respectively. To put this in perspective, MALNET's smallest class contains only 113 samples of the *Click* graph, while 884,455 of the *Adware* type. Models learning from long-tailed distributions tend to favor the majority class, leading to poor generalization performance on rare classes. While class imbalance is traditionally solved by resampling the data (undersampling, oversampling) [12, 13], reshaping the loss function (loss reweighting, regularization) [14, 15] or accounting for input-hardness [16], it is largely unexplored in the graph domain. We hope that MALNET can serve as a source of data to spark novel research in this critical area.

**Cybersecurity Applications.** A majority of malware samples are *polymorphic* in nature, meaning that subtle source code changes in the original malware variant can result in significantly different compiled code (e.g., instruction reordering, branch inversion, register allocation) [17, 18]. Cybercriminals frequently take advantage of this to evade signature based detection, a predominant form of malware detection [19]. Fortunately, these subtle source code changes have minimal effect on the control flow of the executable, which can be represented with a *function call graph* (see Figure 3). Research has demonstrated that function call graphs (FCGs) can effectively defeat the polymorphic nature of malware through techniques like graph matching [20, 21, 22, 23, 24] and representation learning [25, 26]. Unfortunately, prior to the release of MALNET, no large-scale FCG datasets have been made publicly available largely due to the proprietary nature of the data. We note that while open research can significantly advance the frontiers of cybersecurity, it can be used by malicious actors to conduct research on detection avoidance.

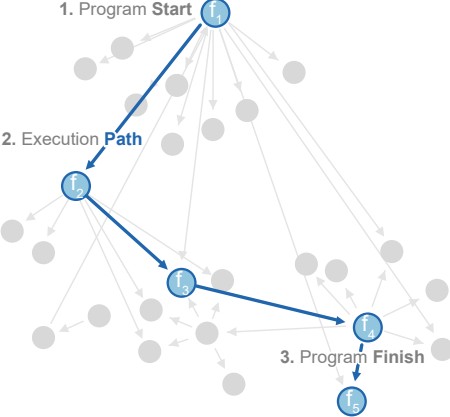

Figure 3: FCG from the *Banker++Trojan* type, and *Acecard* family. Nodes represent functions and edges indicate interprocedural calls. Highlighted in blue is *one* potential execution path.

## 2.1 Graph Representation Learning Databases: Advancing the State-of-the-Art

A number of well labeled small datasets have served as training and evaluation benchmarks for most of today's graph representation learning techniques As the field advances, larger and more challenging datasets are needed for the next generation of algorithms. MALNET offers **105× more graphs**, **39× larger graphs** on average, and **63× the classes**, compared to the popular REDDIT-12K database. We compare MALNET with other graph representation learning datasets and summarize the differences in Table 2, highlighting how MALNET advances the field of graph representation learning by providing large and diverse data.

| Application | Dataset | Graphs | Classes | Ratio | Avg. Node | Avg. Edge | Avg. Degree | Avg. CC | Hierarchy |
|---|---|---|---|---|---|---|---|---|---|
| Cyber | MALNET | 1,262,024 | 696 | 16,901 | 15,378 | 35,167 | 4.34 | .029 | ✓ |
| | CGD [27] | 1,361 | 2 | 1.49 | 782 | 1852 | 4.33 | .095 | - |
| Small molecule | PCBA [28] | 437,929 | 2 | - | 26 | 56 | 2.16 | .002 | - |
| | MUV [29] | 93,087 | 2 | - | 24 | 53 | 2.16 | .001 | - |
| | YEAST [30] | 79,601 | 2 | 1.26 | 22 | 23 | 2.09 | .002 | - |
| | HIV [31] | 41,127 | 2 | - | 26 | 55 | 2.15 | .002 | - |
| | NCI1 [32] | 4,110 | 2 | 1 | 30 | 32 | 2.16 | .003 | - |
| | PTC-MR [33] | 344 | 2 | 1.26 | 14 | 15 | 1.98 | .009 | - |
| | MUTAG [34] | 188 | 2 | 1.98 | 18 | 20 | 2.19 | .000 | - |
| Computer vision | Fingerprint [35] | 2,800 | 4 | 276.5 | 5 | 4 | 1.14 | .001 | - |
| | Letter-low [35] | 2,250 | 15 | 1 | 5 | 3 | 1.32 | .000 | - |
| | Letter-med [35] | 2,250 | 15 | 1 | 5 | 5 | 1.35 | .014 | - |
| | Letter-high [35] | 2,250 | 15 | 1 | 45 | 5 | 1.89 | .298 | - |
| | FIRSTMM-DB [36] | 41 | 11 | 3 | 1377 | 3074 | 4.50 | .263 | - |
| Bioinfo. | DD [37] | 1,178 | 2 | 1.42 | 284 | 716 | 4.98 | .479 | - |
| | PROTEINS [5] | 1,113 | 2 | 1.47 | 39 | 73 | 3.73 | .514 | - |
| | ENZYMES [5] | 600 | 6 | 1 | 33 | 62 | 3.86 | .453 | - |
| Social network | Reddit-T [38] | 203,088 | 2 | 15 | 24 | 12 | 2.01 | .047 | - |
| | Twitch-E [38] | 127,094 | 2 | 1.16 | 30 | 72 | 5.39 | .549 | - |
| | Github-S [38] | 12,725 | 2 | 1.15 | 114 | 117 | 3.19 | .191 | - |
| | REDDIT-12K [6] | 11,929 | 11 | 5.05 | 391 | 457 | 2.28 | .033 | - |
| | Deezer-E [38] | 9,629 | 2 | 1.32 | 23 | 33 | 4.29 | .510 | - |
| | COLLAB [39] | 5,000 | 3 | 3.35 | 74 | 2458 | 37.37 | .891 | - |
| | REDDIT-5K [6] | 4,999 | 5 | 1 | 509 | 595 | 2.25 | .027 | - |
| | REDDIT-B [6] | 2,000 | 2 | 1 | 430 | 498 | 2.34 | .048 | - |
| | IMDB-M [6] | 1,500 | 3 | 1 | 13 | 66 | 8.10 | .969 | - |
| | IMDB-B [6] | 1,000 | 2 | 1 | 20 | 97 | 8.89 | .947 | - |

Table 2: Comparison of MALNET properties with common graph classification datasets. MALNET offers over $1.2$ million graphs averaging $15k$ nodes and $35k$ edges with a hierarchical class structure containing $47$ types and $696$ families. This makes MALNET the largest public graph database constructed to date, offering **105× more graphs**, **39× larger graphs** on average, and **63× more classes** compared to the popular REDDIT-12K database. CC is the clustering coefficient.

**Cybersecurity datasets.** Aside from MALNET, CGD [27] is the only publicly available cybersecurity dataset we could identify for the task of graph classification. In surveying the extensive FCG malware detection literature [20, 25, 21, 26, 22, 23, 24] we observed that almost all data is closed-source; likely due to a combination of security concerns and issues regarding private company data.

**Small molecule datasets.** There are numerous small molecule datasets, including: HIV [31], MUTAG [34], MUV [29], PCBA [28], NCI1 [32], PTC-MR [33], and YEAST [30]. The HIV dataset, introduced by the Drug Therapeutics Program AIDS Antiviral Screen [40], tests the ability of chemical compounds to inhibit HIV replication into one of three classes. MUTAG contains chemical compound graphs divided into two classes according to their mutagenic effect on bacterium. MUV and PCBA are constructed from PubChem BioAssay [41], and contain numerous compounds across 17 tasks and 128 tasks respectively, where each task is a binary classification problem. NCI1 contains chemical compounds, screened for their ability to inhibit the growth of a panel of human tumor cell lines. PTC-MR contains graphs across 2 classes, reporting the effects of chemical compound carcinogenicity on rats. YEAST contains molecule graphs screened for anti-cancer tests, with the binary classification of active or inactive.

**Bioinformatic datasets.** Three popular bioinformatic datasets are: DD [37], ENZYMES [5] and PROTEINS [5]. DD is a data set containing protein structures grouped into 2 categories (enzyme and non-enzyme). ENZYMES contains graphs of protein tertiary enzyme structures with the task of assigning each enzyme to one of 6 levels. Similarly, *PROTEINS* contains protein graphs classified into either enzyme or non-enzyme.

**Computer vision datasets.** Three common computer vision datasets are: Fingerprint [35], FIRSTMM-DB [36] and Letter (low, med, high) [35]. Fingerprint contains fingerprint graphs

across four classes: arch, left, right, and whorl. FIRSTMM-DB contains object point clouds belonging to an object ontology of 11 categories. Letter contains 3 datasets and 15 character classes with varying levels of distortion (low, med, high) added to letter graphs.

**Social network datasets.** Common social network datasets include: COLLAB [39], Deezer Ego-Nets [38], Github Stargazers [38], IMDB (BINARY, MULTI) [42], REDDIT (BINARY, 5K, 12K, Threads) [42, 38] and Twitch Ego-Nets [38]. COLLAB is a collaboration dataset of ego-networks across 3 domains of physics. Deezer Ego-Nets contains user ego-nets across 2 genders from the Deezer music service. Github Stargazers contains graphs of developers who starred either machine learning or web development repositories. IMDB BINARY contains ego-network graphs representing actors and their collaborations across 2 movie genres. IMDB MULTI extends IMDB BINARY with more graphs and 3 movie genres. REDDIT-BINARY contains thread graphs across two content classes (discussion and QA based). REDDIT MULTI-5K contains thread graphs across 5 Reddit thread types. REDDIT MULTI-12K extends REDDIT-5K, containing online discussion thread graphs across 11 classes. REDDIT Threads contains thread graphs across 2 graph classes (discussion, non-discussion). Twitch Ego-Nets contains ego graphs across 2 classes of Twitch users.

# 3 Constructing MalNet

## 3.1 Collecting Candidate Graphs

The first step in constructing MALNET was to identify a source of graph containing the desired properties outlined in Section 2. We determined that the natural abundance, large graph size, and class diversity provided by function call graphs (FCGs) make them an ideal source of graphs. While FCGs, which represent the control flow of programs (see Figure 3), can be statically extracted from many types of software (e.g., EXE, PE, APK), we use the Android ecosystem due to its large market share [43], easy accessibility [44] and diversity of malicious software [45]. With the generous permission of the AndroZoo repository [46, 44], we collected 1,262,024 Android APK files, specifically selecting APKs containing both a *family* and *type* label obtained from the Euphony classification structure [8]. This process took about a week to download and 10TB in storage space when using the maximum allowed 40 concurrent downloads. In addition, we spent about 1 month collecting raw VirusTotal (VT) reports to release with MALNET, through VT's academic access, which allows 20k queries per day. Each VT report contains up to 70 antivirus labels per graph.

## 3.2 Processing the Graphs

Once the APK files and labels were collected, we extract the function call graphs by running the files through Androguard [47], which statically analyzes the APK's DEX file. Distributed across Google Clouds General-purpose (N2) machine with 16 cores running 24 hours a day, the process took about 1 week to extract the graphs. We leave each graph in its original state—retaining its edge directionality, disconnected components and node isolates (i.e., single nodes with no incident edges). On average, each graph has $15,378$ nodes and $35,167$ edges; and typically contains a single giant connected component, many small disconnected components, and numerous node isolates. Table 1 describes the 10 graph types (out of 47) that have the highest number of graphs. Appendix Table 4 provides a full analysis on all graph type. Each graph is stored in a standard edge list format for its wide support, readability, and ease of use. In total, the graphs' edge list files consume over 443 GB of hard disk space. Since we are dealing with highly malicious software, our goal is to mitigate the risk of releasing information that could potentially be used to reverse engineer malware. Thus, we numerically relabel the nodes of each graph, removing any associated attribute information, which makes reverse engineering highly unlikely. However, malicious actors could develop new variants of detection-resistant malware that looks structurally similar to benign function call graphs, by gleaning graph structure knowledge from MALNET in the absence of node and edge labels.

## 3.3 MalNet-Tiny

We construct MALNET-TINY, containing $5,000$ graphs across balanced 5 *types*. In addition, we limit each graph to contain at most 5k nodes so that the dataset is truly "tiny". The goal of MALNET-TINY is to enable users to rapidly prototype new ideas, since it requires only a fraction of the time needed to train a new model. MALNET-TINY is released alongside the full dataset at `https://mal-net.org`.

## 3.4 Online Exploration of the Data

To assist researchers and practitioners in exploring MALNET, we have designed and developed MALNET EXPLORER, an interactive graph exploration and visualization tool. It runs on most modern web browsers (Chrome, Firefox, Safari, and Edge), platforms (Windows, Mac OS, Linux), and devices (Android and iOS). Our goal is to enable users to easily explore the data before downloading. MALNET EXPLORER's user interface uses a *responsive* design that automatically adjusts its component layout, based on the users' device types and screen resolutions. MALNET EXPLORER is available online at: `https://mal-net.org`.

## 4 MalNet for New Research & Discoveries

MALNET is substantially larger than existing graph databases used for graph representation learning research, with many more graphs, much larger graphs, and many more classes of graphs. Such unprecedented advancements provides exciting opportunities to make new discoveries and explore new research directions previously not possible. In this section, we present our findings to demonstrate such possibilities. We discuss the experimental setup below, followed by an overview of the graph representation techniques in Section 4.1. Section 4.2 discusses the new discoveries we found by studying MALNET; and Section 4.3 highlights new research directions enabled by MALNET.

**Experimental Setup.** We divide MALNET into three stratified sets of data: training, validation and test, with a split of 70/10/20, respectively; repeated for graph *type*, *family* and MALNET-TINY labels. Each model is evaluated on its macro-F1 score, however, we report three performance metrics—macro-F1, precision and recall, as is typical for highly imbalanced datasets [16, 48]. We perform our experiments in Python3 using a DGX A-100 containing 128 CPU cores and 8 A-100 GPUs.

### 4.1 Graph Representation Techniques

We present results for 7 strong, recent, scalable, and readily available graph representation techniques [49, 38]. Specifically, we evaluate 2 graph neural network (GNN) models [50, 51] and 5 data mining techniques [1, 3, 52, 53]. We leave the graph in its natural state for each GNN i.e., directed graph with isolates; and follow recommended preprocessing steps from the paper of each data mining technique. In addition, each data mining embedding techniques uses a random forest model for the task of graph classification, where we run a grid search across the validation set to identify the number of estimators $n_e \in [1, 5, 10, 25, 50]$ and tree depth $t_d \in [1, 5, 10, 20]$. All hyperparameters are individually tuned for *type*, *family* and *tiny* classification levels. We briefly summarize each method and its configuration below:

1. **GCN** [50] is a graph neural network which learns network embeddings by aggregating node features over neighborhoods. Following [51], we use 5 GNN layers and an Adam optimizer [54]. We set node features using LDP [1], and add self loops which has been shown to improve performance [55]. We tune hyperparameters for (1) the number of hidden units $\in \{32, 64\}$ and (2) the learning rate $\in \{0.001, 0.0001\}$, repeated for both *type* and *family* classification levels. We find that 64 units with a learning rate of 0.0001 performs best. Running this search took over **26 days** using the Nvidia DGX A100, their most powerful commercial GPU server.

2. **GIN** [51] is a state-of-the-art GNN with strong theoretical backing. Following [51], we set $\epsilon = 0$, use 5 GNN layers, and an Adam optmizer [54]. We set node features using LDP [1], and add self loops which has been shown to improve performance [55]. We tune hyperparameters for (1) the number of hidden units $\in \{32, 64\}$ and (2) the learning rate $\in \{0.001, 0.0001\}$. We find that 64 units with a learning rate of 0.0001 performs best. Running this search took over **23 days** using the Nvidia DGX A100.

3. **LDP** [1] is a simple representation scheme that summarizes each node and its 1-hop neighborhood using using 5 degree statistics. These node features are then aggregated into a histogram where they are concatenated into feature vectors. We use the parameters suggested in [1]. Running this method took 4 hours parallelized across all 128 CPU cores of the Nvidia DGX A100.

4. **NoG** [3] ignores the topological graph structure, viewing the graph as a two-dimensional feature vector of the node and edge count. Running this method took approximately 1 hour parallelized across all 128 CPU cores of the Nvidia DGX A100.

| Method | Type | | | Family | | | TINY |
|---|---|---|---|---|---|---|---|
| | Macro-F1 | Precision | Recall | Macro-F1 | Precision | Recall | Accuracy |
| Feather [52] | **.41** | .71 | .35 | **.34** | .56 | .29 | .86 |
| LDP [1] | .38 | .69 | .31 | **.34** | .55 | .28 | .86 |
| GIN [51] | .39 | .57 | .36 | .28 | .32 | .28 | **.90** |
| GCN [50] | .38 | .51 | .35 | .21 | .24 | .21 | .81 |
| Slaq-LSD [53] | .33 | .62 | .26 | .24 | .42 | .19 | .76 |
| NoG [3] | .30 | .62 | .25 | .25 | .42 | .21 | .77 |
| Slaq-VNGE [53] | .04 | .07 | .04 | .01 | .01 | .01 | .53 |

Table 3: Comparison of macro-F1, precision and recall scores achieved by 7 methods at the *type* (low diversity, with 47 classes) and *family* (high diversity, with 696 classes) and *tiny* (5k graphs across 5 balanced classes) classification levels. Comparing methods across *type* and *family*, the classification task becomes increasingly difficult as diversity and data imbalance increase.

5. **Feather** [52] is a more complex representation scheme that uses characteristic functions of node features with random walk weights to describe node neighborhoods. We perform a search over the key *order* $\in \{4, 5, 6\}$parameter, which controls how much information is seen from higher order neighborhoods. We find that an order of 5 performs best. For the remaining parameters, we use the values suggested in [52]. Running this search took over 19 hours parallelized across all 128 CPU cores of the Nvidia DGX A100.

6. **Slaq-VNGE** [53] approximates the spectral distances between graphs based on the Von Neumann Graph Entropy (VNGE), which measures information divergence and distance between graphs [56]. We perform a search over 2 key parameters: number of random vectors $n_v \in \{10, 15, 20\}$ and the number of Lanczos steps $s \in \{10, 15, 20\}$. We find that $n_v = 15$ and $s = 15$ performs best. For the remaining parameters, we use the values suggested in [53]. Running this search took 8 hours parallelized across all 128 CPU cores of the Nvidia DGX A100.

7. **Slaq-LSD** [53] approximates NetLSD, which measures the spectral distance between graphs based on the heat kernel [57]. We perform a search over 2 key parameters: number of random vectors $n_v \in \{10, 15, 20\}$ and number of Lanczos steps $s \in \{10, 15, 20\}$. We find that $n_v = 20$ and $s = 20$ performs best. For the remaining parameters, we use the values suggested in [53]. Running this search took 8 hours parallelized across all 128 CPU cores of the Nvidia DGX A100.

**Limitations.** We tested a number of alternative graph representation techniques and decided to exclude them—methods based on kernal [58, 59, 60, 60], spectral [61, 57, 62, 63, 64] and document embedding [65, 66]—as they were computationally prohibitive for the scale of MALNET, making it infeasible to run the techniques over the full dataset or perform parameter selection. We also note that methods that work well on other datasets may not work well on MALNET due to its larger scale and different structural properties (see Table 2); vice-versa, methods that work on MALNET may not transfer well to other datasets. We hope MALNET will inspire the release of additional large-scale datasets in the call graph domain and other novel application areas, which will help enable researchers to develop and evaluate methods that generalize across domains.

## 4.2 Enabling New Discoveries

Current graph representation research uses datasets that are significantly smaller in scale, and much less diverse compared to MALNET. In light of this, we want to study what new discoveries can be made, that were previously not possible due to dataset limitations. For example, what is the impact of class imbalance and diversity in the classification process? We synthesized our findings into the following 2 major discoveries (D1-D2).

**D1. Less Diversity, Better Performance.** Comparing methods in Table 3 across malware *type* (low diversity, with 47 classes) and *family* (high diversity, with 696 classes), the classification task becomes increasingly difficult as diversity and data imbalance increase. This trend is visible across all 7 graph representation methods. For the best performing method, Feather, the

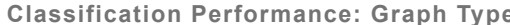

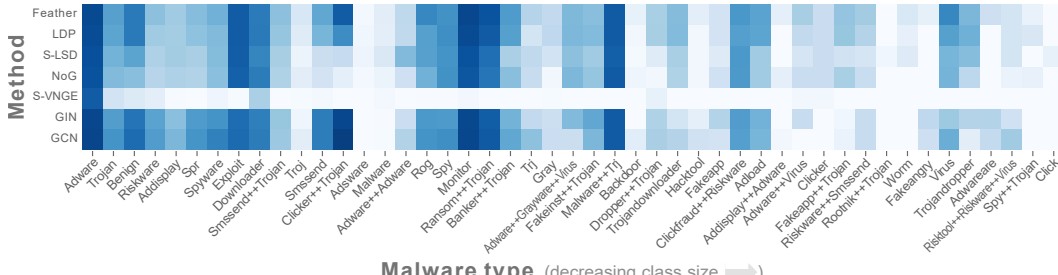

Figure 4: Class-wise comparison of model predictions where a darker cell represents a higher F1 score. We observe that certain classes are more challenging to classify than others.

macro-F1 score drops from $0.41$ (type) to $0.34$ (family). This matches our intuition from the "tiny" experiments in Table 3, which shows strong method performance when evaluating on a small subset of MALNET, containing 5 well-balanced types.

**D2. Simple Baselines Surprisingly Effective.** Both NoG and LDP use basic graph statistics. Given the simplicity of these methods, they perform remarkably well, often outperforming or matching the performance of more complex methods. For example, in Table 3 we can see that LDP ties for the best performing *family* classification method, achieving a macro-F1 score of $0.34$, while beating significantly more complex methods e.g., GIN, GCN, Slaq-LSD. A similar trend is found in *type* level classification results where LDP outperforms SLAQ-LSD and performs on par with GIN and GCN, despite being simpler and significantly faster than all 3 methods. Using small graph databases, earlier work [1] suggested the potential merits of considering simpler approaches. For the first time, using the largest graph database to date, our result confirms that many current techniques in the literature do not well capture non-attributed graph topology.

## 4.3 Enabling New Research Directions

The unprecedented scale and diversity of MALNET opens up new exciting research opportunities for the graph representation community. Below, we present four promising directions (R1-R4).

**R1. Class Hardness Exploration.** Because of MALNET's large diversity, it is now possible for researchers to explore why certain classes are more challenging to classify than others. For example, Figure 4 shows *Malware++Trj* significantly outperforming both *Troj* and *Adsware*, which contain many more examples. This result is surprising, and provides strong impetus for additional research into class hardness, such as: (a) investigating whether existing methods are flexible enough to represent the diverse graph structures; and (b) inviting researchers to study the similarities across class types (e.g. merge *Spr* and *Spyware*). To support further development in this challenging area, we release the raw VirusTotal reports containing up to 70 labels per graph.

**R2. Imbalanced Classification Research.** The natural world often follows a long-tailed data distribution where only a few classes account for most of the examples [16]. As evidenced in discovery **D1**, the long-tail often causes classifiers to perform well on the majority class, but poorly on rare ones. Unfortunately, imbalanced classification research in the graph domain has yet to receive much attention, largely because no datasets existed to support the research. By releasing MALNET, the largest naturally imbalanced database to date, we hope foster new interest in this important area.

**R3. Reconsidering Merits of Simpler Graph Classification Approaches.** Our discovery in **D2** indicates that simpler methods can match or outperform more recent and sophisticated techniques, suggesting that current techniques aiming to capture graph topology are not yet well-reflected for non-attributed graphs, echoing results from [1]. More broadly, our discovery demonstrates—for the first time—such phenomenon at the unprecedented scale and diversity offered by MALNET. We believe our results will inspire researchers to reconsider the merits of simpler approaches and classic techniques, and to build on them to reap their benefits.

**R4. Enabling Explainable Research.** In Figure 4, we observe that certain representation techniques better capture particular graph types. For example, Feather, GIN and GCN significantly outperforms other methods on *Clicker++Trojan*. This is an interesting result, as it could provide insight into when one technique is preferred over another (e.g., local neighborhood structure, global graph structure, graph motifs). We believe that the wide range of graph topology and substructures contained in MALNET's nearly 700 classes will enable new explainability research.

# 5 Conclusion

The study of graph representation learning is a critical tool in the characterization and understanding of complex interconnected systems. Currently, no large-scale database exists to accurately assess the strengths and weaknesses of these techniques. To address this, we contribute a new large-scale database—MALNET—containing $1,262,024$ graphs, averaging over 15k nodes and 35k edges per graph, across a hierarchy of $47$ types and $696$ families. We hope MALNET will become a central resource for a broad range of graph research. The database is available at `www.mal-net.org`.

# 6 Acknowledgements

We thank Kevin Allix and AndroZoo colleagues for generously allowing us to use their data in this research; this work was in part supported by NSF grant IIS-1563816, CNS-1704701, GRFP (DGE-1650044), a Raytheon research fellowship, and an IBM PhD fellowship.

# Appendix

| Type | # graphs | # fams. | Nodes | | | | Edges | | | | Avg. Degree | | | |
|---|---|---|---|---|---|---|---|---|---|---|---|---|---|---|
| | | | min | mean | max | std | min | mean | max | std | min | mean | max | std |
| Adware | 884K | 250 | 7 | 14K | 211K | 16K | 4 | 31K | 605K | 38K | 0.50 | 2.21 | 6.24 | 0.36 |
| Trojan | 179K | 441 | 5 | 15K | 228K | 18K | 4 | 34K | 530K | 42K | 0.58 | 2.05 | 6.74 | 0.52 |
| Benign | 79K | 1 | 5 | 35K | 552K | 30K | 3 | 79K | 2M | 74K | 0.58 | 2.13 | 5.30 | 0.31 |
| Riskware | 32K | 107 | 5 | 12K | 173K | 16K | 4 | 30K | 334K | 39K | 0.58 | 2.16 | 5.42 | 0.56 |
| Addisplay | 17K | 38 | 37 | 13K | 98K | 15K | 37 | 28K | 246K | 34K | 0.92 | 1.97 | 4.38 | 0.37 |
| Spr | 14K | 46 | 12 | 28K | 169K | 21K | 7 | 67K | 369K | 52K | 0.58 | 2.27 | 4.70 | 0.44 |
| Spyware | 7K | 19 | 12 | 5K | 55K | 6K | 7 | 11K | 121K | 14K | 0.58 | 1.95 | 4.27 | 0.46 |
| Exploit | 6K | 13 | 19 | 24K | 102K | 14K | 14 | 45K | 250K | 30K | 0.74 | 1.88 | 3.34 | 0.33 |
| Downloader | 5K | 7 | 37 | 20K | 107K | 28K | 37 | 46K | 321K | 63K | 0.96 | 1.68 | 3.53 | 0.66 |
| Smssend++Trojan | 4K | 25 | 16 | 34K | 147K | 19K | 13 | 82K | 387K | 48K | 0.81 | 2.39 | 3.78 | 0.23 |
| Troj | 3K | 36 | 14 | 6K | 64K | 8K | 11 | 15K | 115K | 18K | 0.79 | 1.98 | 5.60 | 0.52 |
| Smssend | 3K | 12 | 15 | 20K | 111K | 14K | 12 | 49K | 337K | 38K | 0.80 | 2.34 | 4.61 | 0.47 |
| Clicker++Trojan | 3K | 3 | 220 | 6K | 29K | 3K | 471 | 14K | 72K | 7K | 1.52 | 2.33 | 2.92 | 0.18 |
| Adsware | 3K | 16 | 368 | 11K | 53K | 13K | 564 | 26K | 143K | 28K | 1.02 | 2.19 | 4.27 | 0.26 |
| Malware | 3K | 19 | 6 | 8K | 119K | 13K | 5 | 16K | 286K | 29K | 0.83 | 1.90 | 3.97 | 0.67 |
| Adware++Adware | 3K | 2 | 192 | 9K | 55K | 6K | 289 | 20K | 138K | 16K | 1.49 | 2.16 | 3.17 | 0.27 |
| Rog | 2K | 22 | 26 | 15K | 102K | 19K | 31 | 35K | 232K | 46K | 0.91 | 2.05 | 4.79 | 0.49 |
| Spy | 2K | 7 | 48 | 22K | 107K | 15K | 44 | 49K | 271K | 40K | 0.92 | 2.17 | 3.07 | 0.25 |
| Monitor | 1K | 5 | 329 | 4K | 41K | 5K | 580 | 7K | 102K | 12K | 1.53 | 1.83 | 3.09 | 0.21 |
| Ransom++Trojan | 1K | 7 | 556 | 51K | 139K | 22K | 965 | 115K | 319K | 48K | 1.59 | 2.26 | 2.59 | 0.21 |
| Banker++Trojan | 1K | 6 | 29 | 33K | 103K | 16K | 36 | 72K | 237K | 38K | 1.22 | 2.15 | 2.99 | 0.24 |
| Trj | 940 | 18 | 29 | 13K | 171K | 16K | 36 | 30K | 402K | 39K | 1.15 | 2.20 | 4.44 | 0.49 |
| Gray | 922 | 10 | 51 | 16K | 66K | 13K | 56 | 39K | 153K | 31K | 0.88 | 2.09 | 4.33 | 0.58 |
| Adware++Grayware++Virus | 835 | 4 | 22 | 6K | 84K | 13K | 20 | 14K | 193K | 29K | 0.86 | 2.79 | 3.17 | 0.34 |
| Fakeinst++Trojan | 718 | 10 | 51 | 15K | 94K | 17K | 58 | 37K | 229K | 44K | 0.99 | 2.12 | 2.84 | 0.48 |
| Malware++Trj | 609 | 1 | 52K | 52K | 56K | 596 | 118K | 119K | 128K | 1K | 2.28 | 2.28 | 2.29 | 0 |
| Backdoor | 602 | 10 | 25 | 13K | 146K | 22K | 21 | 33K | 427K | 57K | 0.84 | 2.19 | 3.55 | 0.37 |
| Dropper++Trojan | 592 | 8 | 47 | 5K | 67K | 7K | 50 | 11K | 175K | 18K | 1.06 | 1.98 | 3.92 | 0.70 |
| Trojandownloader | 568 | 7 | 1K | 38K | 102K | 19K | 2K | 86K | 258K | 45K | 1.34 | 2.19 | 2.54 | 0.21 |
| Hacktool | 542 | 7 | 668 | 17K | 41K | 9K | 2K | 37K | 92K | 20K | 1.63 | 2.21 | 3.64 | 0.25 |
| Fakeapp | 425 | 5 | 24 | 4K | 50K | 7K | 21 | 8K | 107K | 16K | 0.88 | 1.67 | 2.79 | 0.37 |
| Clickfraud++Riskware | 369 | 5 | 2K | 18K | 20K | 2K | 4K | 38K | 43K | 5K | 1.95 | 2.13 | 2.25 | 0.04 |
| Adload | 333 | 4 | 2K | 19K | 53K | 18K | 4K | 48K | 149K | 48K | 1.46 | 2.29 | 3.13 | 0.40 |
| Addisplay++Adware | 294 | 1 | 3K | 20K | 50K | 9K | 6K | 41K | 108K | 20K | 1.65 | 2.03 | 2.45 | 0.21 |
| Adware++Virus | 274 | 9 | 38 | 15K | 59K | 15K | 38 | 33K | 138K | 35K | 1 | 2.22 | 3.17 | 0.54 |
| Clicker | 265 | 5 | 47 | 3K | 75K | 7K | 43 | 6K | 190K | 17K | 0.91 | 1.62 | 3.32 | 0.51 |
| Fakeapp++Trojan | 256 | 1 | 44 | 21K | 72K | 15K | 39 | 41K | 162K | 34K | 0.88 | 1.74 | 2.30 | 0.27 |
| Riskware++Smssend | 247 | 7 | 12 | 2K | 60K | 6K | 7 | 5K | 154K | 14K | 0.58 | 1.68 | 3 | 0.45 |
| Rootnik++Trojan | 223 | 5 | 210 | 16K | 84K | 21K | 395 | 39K | 197K | 46K | 1.15 | 2.59 | 3.21 | 0.47 |
| Worm | 220 | 7 | 64 | 14K | 94K | 15K | 78 | 31K | 204K | 34K | 0.99 | 1.99 | 3.42 | 0.40 |
| Fakeangry | 211 | 2 | 516 | 6K | 98K | 11K | 946 | 15K | 279K | 29K | 1.70 | 2.35 | 3.29 | 0.27 |
| Virus | 191 | 3 | 681 | 15K | 80K | 19K | 1K | 35K | 177K | 46K | 1.32 | 2.12 | 3.18 | 0.33 |
| Trojandropper | 178 | 4 | 220 | 20K | 78K | 18K | 236 | 39K | 185K | 39K | 1.03 | 1.83 | 4.36 | 0.32 |
| Adwareare | 152 | 3 | 893 | 26K | 57K | 14K | 2K | 60K | 144K | 32K | 1.88 | 2.25 | 2.60 | 0.20 |
| Risktool++Riskware++Virus | 152 | 3 | 37 | 16K | 65K | 16K | 37 | 36K | 158K | 37K | 1 | 1.92 | 3.17 | 0.48 |
| Spy++Trojan | 119 | 5 | 54 | 31K | 118K | 25K | 66 | 75K | 293K | 61K | 1.22 | 2.31 | 3.26 | 0.37 |
| Click | 113 | 1 | 2K | 4K | 12K | 2K | 4K | 8K | 26K | 4K | 1.80 | 2.04 | 2.74 | 0.21 |

Table 4: Descriptive statistics for each graph type in MALNET.

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
