# OpenReview forum: "A Large-Scale Database for Graph Representation Learning"
_NeurIPS.cc/2021/Track/Datasets_and_Benchmarks/Round1 — NeurIPS 2021 Datasets and Benchmarks Track (Round 1)_

### Official Review · Reviewer_JwuZ · 2021-06-28
**Great new dataset for graph representations**

**Rating:** 9
**Confidence:** 5
**Clarity:** This paper is exceptionally well-writ…

**Strengths:**

The paper greatly expands the data available for graph representation learning. The dataset is very rich, containing great variety in graph type and biases that are currently unavailable for research datasets.

The manner in which the dataset was produced is straightforward, and it should be conceptually simple to extend this sort of analysis to other codebases.

There are a number of potential domain-specific applications, including computer security, and program analysis and refactoring.

The exploration of various graph representation techniques suggests several interesting directions for future research, including that the predictive power of very simple graph statistics (like node and edge counts) may be underestimated in the current literature.

The authors provide an easy-to-use website for exploring the dataset, with engaging visual representations of program call graphs. Overall, I am excited about this paper, and would be interested in exploring this data in my own research.

**Weaknesses:**

No major noteworthy weaknesses.

**Additional Feedback:**

Just a couple of small suggestions that would be very nice to have, but are not essential for the publication of this paper:
* A brief comparison of the qualitative properties of these call graphs, and how they might differ from other datasets (e.g. number of connected components, density, etc.)
* A bit of exposition on the properties we might expect various types and family of call graph to have based on prior knowledge of computer security research.

**Correctness:**

The dataset is constructed in a clear and straightforward manner. The authors train a suite of standard graph learning algorithms on this data to get a sense of performance, including GCNs, LDPs and others. When the authors did not test a standard method (such as NLP-based string embedding methods), they explain clearly the performance bottlenecks preventing them from doing so.

**Documentation:**

The dataset is well-documented, and download links for reviewers and researchers to use the data are already available and straightforward to use.

In addition, the authors took great care to obfuscate certain aspects of the original data (in particular, the method names for functions in the call graph), in order to mitigate the potential for the malware used to form these call graphs to be reproduced using this research. The authors carefully considered the security implications of this work.

**Ethics:**

The authors directly and satisfactorily addressed the main potential ethical concern for this work, namely the potential security implications of studying malware.

**Relation To Prior Work:**

The authors systematically detail previous datasets for graph representation learning, how the dataset in this paper differs from those, and the kinds of new research their dataset will enable.

**Summary And Contributions:**

The authors introduce MalNet, a new labelled graph dataset containing over 1 million new function call graphs produced from various kinds of function call graphs found in malicious programs and software. The graphs are labelled into type and family of malware. (Examples of malware types include "Adware", "Trojan" and "Spyware.") The labels are generated from antivirus companies. These call graphs are non-trivial in size, containing an average of thousands of nodes and edges per graph.

---

> ### Author Response · Authors · 2021-07-12
> **Response to Reviewer JwuZ**
>
> Thank you for your encouraging comments and strong support. It’s great to hear that you are excited about exploring MalNet in your own research.
>
> You make a great point about analyzing and comparing the structural properties of MalNet to other graph representation learning datasets. We updated Table 2 to include statistics for average degree and average clustering coefficient. Thanks for your suggestion to further explore call graph properties of malware families and types; this is a great future work direction.

---

> > ### Comment · Reviewer_JwuZ · 2021-07-20
> > **Followup**
> >
> > Thank you! I look forward to reading about the followup work.

---

### Official Review · Reviewer_4Npm · 2021-07-04
**Graph representation learning dataset for security applications; comprehensive and well-written**

**Rating:** 8
**Confidence:** 3
**Correctness:** The database seems soundly constructe…

**Strengths:**

The paper is strong in the database collected, in its exposition, and in its testing of the dataset. In particular, the online database linked in the paper is quite comprehensive for visualization and testing, and shows interesting applications in cybersecurity.

**Weaknesses:**

One concern I have about the study is that this kind of database can also be used for 'bad', e.g. for training malware algs. The authors mention that they numerically relabel the nodes of each graph in order to mitigate potential reverse engineering, but a more in-depth discussion about the potential would strengthen the paper. (The authors also mention this is basically the only such open database).

One part that was confusing to me was in section 2, the explanation for different applications: while the cybersecurity explanation is clear, the other ones just seem to mention existing datasets in other fields, raising the question of how is this dataset adding to those applications, or is that just related work?

**Additional Feedback:**

--

**Clarity:**

The paper is clearly written, only two minor details:  lines 34-35 are interrupted and line 225 says ‘optmizer'

**Documentation:**

The paper presents a url at which the data can be found, lying on a comprehensive website with explanations, visualizations, and applications.

**Ethics:**

The ethical concern being raised is how this dataset could be used for malware applications (the authors do mention that graph nodes are relabelled for that purpose, but more discussion would help).

**Relation To Prior Work:**

The paper does a good job of relating to related datasets.

**Summary And Contributions:**

This paper contributes a database for graph representation, Malnet, containing 1.2 million graphs from different classes. The database is comprehensive and has important cybersecurity applications, as the authors describe. The authors use the AndroZoo repository in collecting the Android APK files, extracting then the function call graphs, and using them for different graph representation techniques and model predictions. Two interesting contributions of this dataset are: showing that current techniques in the literature don't capture non-attributed graph topology and showing that diversity inversely correlates with performance (to an extend, intuitive).

---

> ### Author Response · Authors · 2021-07-12
> **Response to Reviewer 4Npm**
>
> We appreciate your thoughtful feedback. We have improved our paper based on your suggestions, as follows.
>
> The potential misuse of MalNet is something we took into careful consideration when constructing the database. Thank you for suggesting a more in-depth discussion. We have revised two parts to strengthen the paper: (1) We updated Sec 3.2 to discuss the possibility that malicious actors could develop new variants of detection-resistant malware that looks structurally similar to benign function call graphs, by gleaning graph structure knowledge from MalNet in the absence of node and edge labels. (2) We updated Sec 2 (“Cybersecurity Applications” subsection) to discuss how open research in improving malware detection could potentially lead to bad actors conducting research on avoiding detection, as Reviewer N7oL commented.
>
> Thank you for your clarifying question on the description of other existing datasets. We revised the title of Sec 2.1, and updated the last sentence of the first paragraph to clearly state that our goal is to compare MalNet with other graph representation learning datasets, and to highlight how MalNet advances the field of graph representation learning by providing large and diverse data.
>
> Thanks for catching the typos. We have fixed them!

---

> > ### Comment · Reviewer_4Npm · 2021-07-13
> > **Response to comment**
> >
> > Thank you, authors! That sounds good to me, I appreciate you taking the time to develop the sections, particularly in the cybersecurity applications.

---

### Official Review · Reviewer_N7oL · 2021-07-06
**Review (updated 19th July 2021)**

**Rating:** 7
**Confidence:** 3
**Correctness:** Seems good

**Strengths:**

- clearly identify shortcomings in existing datasets
- graph representation is of interest in several application domains so the new dataset can have a significant impact
- hosting via the Georgia Institute of Technology serves provides accessibility
- tests of existing methods as well as proposal of new research directions

**Weaknesses:**

- graphs from different domains can have potentially very different structures, while this is a very large dataset its data comes from a very specific domain
- potential limitations of this limited domain aren't discussed

**Additional Feedback:**

The content of Section 2.1. and Table 2 is very redundant. Instead of just restating statistics the space would be used better on describing the differences in the tasks and structures.

Especially interesting would be a discussion of the potential limitations of this dataset given its specific domain.

Section 4.1. You mention the time it takes to train the complicated methods, but not the simple ones. For a better comparison, all training times would be interesting.

**Clarity:**

Overall the paper is well written and clearly structured. One small exception, it's not very clearly stated that the dataset is specifically about functions calls from malware. It is in the name of the dataset, but not really in the text.


**Documentation:**

Yes, the paper clearly describes the collection process, the supplementary material gives additional information and the website with the dataset is online.

**Ethics:**

No. The dataset is about malware but only the function call graphs which themselves are benign. Research on this dataset can potentially improve malware detection. However, as with any such topic, open research in better detection can also potentially be used for research on how to avoid detection.

**Relation To Prior Work:**

Existing datasets and their limitations are discussed.

**Summary And Contributions:**

The paper provides a new database for graph representation learning. The graphs in the database were generated from function call graphs of malware software. Both the number of graphs and the size of the graphs within the database are significantly larger than existing databases for graph representation learning. Finally, the authors use this new dataset to test existing graph representation learning approaches and propose new potential research directions on this dataset.

**Update**
There seems to be a consensus for acceptance. I appreciate the author's response to my review.

---

> ### Author Response · Authors · 2021-07-12
> **Response to Reviewer N7oL**
>
> Thank you very much for your thoughtful feedback and suggestions for improvements. We have addressed your concerns based on your comments.
>
> We agree with you that different domains can potentially contain very different graph structures. To address this, we updated Table 2 to include statistics for average degree and average clustering coefficient (we’ll also investigate other interesting statistics to include). We believe this will help readers better understand the properties of working with various datasets in different domains.
>
> Thank you for your suggestion for reducing redundancy between the text in Sec 2.1 and Table 2. We have removed the majority of graph statistics from the text so readers can better focus on the differences in tasks across datasets.
>
> Thanks for spotting that we did not clearly state that this dataset is focused on malicious software function call graphs. We updated the abstract and last paragraph of the introduction (first sentence) to clearly state this fact.
>
> Following your advice, we revised Sec 4.1 to add the time it takes to train all the methods.
>
> Thank you for suggesting that we include a discussion of the potential limitations. We have revised the last paragraph in Section 4.1 to explicitly discuss the limitations. Specifically, we point out that methods that work well on MalNet may not transfer to other datasets, and vice versa. We hope our work will inspire the release of additional large-scale datasets in the call graph domain and other novel application areas, which will help enable researchers to develop and evaluate methods that generalize across domains. In addition, we talk about the requirements that methods be scalable to work with MalNet.

---

### Comment · Reviewer_P2xb · 2021-08-06
**An official ethics review**

This is a comment from an official ethics reviewer.

The ethical concern raised by the reviewers was on misuse. The authors have structured the dataset appropriately and added sufficient discussion to the paper. There is no reason to hold the paper back for ethical concerns.

---

### Decision · Program_Chairs · 2021-07-26

Accept